# Independent Double-Boost Interleaved Converter with Three-Level Output

**Vasile Mihai Suciu, Sorin Ionut Salcu, Alexandru Madalin Pacuraru, Lucian Nicolae Pintilie, Norbert Csaba Szekely and Petre Dorel Teodosescu ***

Department of Electrical Machines and Drives, Technical University of Cluj-Napoca, 400489 Cluj-Napoca, Romania; mihai.suciu@emd.utcluj.ro (V.M.S.); sorin.salcu@emd.utcluj.ro (S.I.S.); alexandru.pacuraru@mae.utcluj.ro (A.M.P.); lucian.pintilie@emd.utcluj.ro (L.N.P.); norbert.szekely@emd.utcluj.ro (N.C.S.)
* Correspondence: petre.teodosescu@emd.utcluj.ro

**Abstract:** This paper introduces a novel converter topology based on an independent controlled double-boost configuration. The structure was achieved by combining two independent classic boost converters connected in parallel at the input and in series at the output. Through proper control of the two boost converters, an interleaved topology was obtained, which presents a low ripple for the input current. Being connected in series at the output, a three-level structure was attained with twice the voltage gain of classic boost and interleaved topologies. A significant feature of the proposed converter is the possibility of independent operation of the two integrated boost converters, in both symmetrical and asymmetrical modes. This feature may be particularly useful in voltage balancing or interconnection with bipolar DC grids/applications. The operation principle, simulations, mathematical analysis, and laboratory prototype experimental results are presented.

**Keywords:** double-boost converter; interleaved; three-level voltage; renewable energy; voltage gain

## 1. Introduction

The need for high-gain boost topology converters with a low ripple of the input current can be justified in various applications, such as bidirectional chargers for electric vehicles in smart homes [1], development of energy storage for renewable power systems [2–4], and DC microgrid and mobility-related concepts [5–8]. The present work focuses on the power conversion stage of low voltage DC renewable sources, such as photovoltaic panels with power optimizers and/or micro-inverter applications in AC or DC distribution microgrids [9].

The classic interleaved boost converter is recognized for its high efficiency and low ripple input current. On the other hand, this type of converter exhibits a limited voltage gain. High-gain interleaved DC–DC voltage converters in transformerless structures have been explored in various studies [10–13]. A floating, high-gain voltage interleaved converter widely employed in research studies was analyzed in [14,15]. This structure is based on the principle of the intercalated operation of two classic boost converters. However, the output circuit for this application does not have purely capacitive filtering, thus it requires a more complex scheme for the output voltage control. Numerous studies commonly use the three-level boost converter [16], but since the input source and inductor currents are the same, there is no interleaved characteristic, and the voltage gain is limited. The same limitations were found in the research of [17], where a three-level DC–DC boost converter with dual output was presented. An interleaved step-up DC–DC converter topology composed of two boost converters was introduced in [18]. During operation, the converter stages are indirectly connected in a series, which ensures a high voltage gain of the output. Since the output consists of two series capacitors, a three-level output is achieved. The individual voltage drop across each capacitor is balanced according to the operating state of the converter, but a fully independent control is not achieved for

this approach, and the efficiency at a low input voltage is limited. The same limitation regarding the independent voltage balancing of the output capacitors was found in [19].

High voltage gain non-isolated converters without coupled inductors exhibit a simpler structure, being more suited for low power level applications [20–31]. In Ref. [20], high conversion gain and efficiency were achieved using an interleaved topology. To acquire high voltage gain with low voltage stress across the semiconductor devices, [21] proposed the usage of an active–passive inductor cell configuration for a step-up DC–DC converter, and [22] showed that the implementation of a switched capacitor topology allowed a wide voltage gain range. In Refs. [23–26], multistage diode–capacitor (MSDC) networks were implemented. Similarly, in Ref. [27], a switched inductor and switched capacitor multilevel topology was involved, with the output voltage distributed among multiple levels of output capacitors, subsequently reducing the voltage stress on the components. Using two duty ratios to operate the switched states of the converter, Ref. [28] attained high voltage gain without using extreme values of these duty ratios, thus the voltage conversion proved to be more efficient. In addition, in Refs. [29–31], the converter topologies employed an interleaved design, with the inductors switched in a parallel and series configuration during the charging and discharging states. Despite the high voltage gain, in all the variants mentioned above, there was high voltage stress across the power semiconductors. All these converter topologies have good gain and efficiency, and some also have interleaved properties, but none have a three-level output with fully controlled voltage operation.

The present work introduces a novel interleaved converter topology that integrates two classic boost converters working independently, in parallel at the input and in series at the output. The circuit has high voltage gain capabilities, and in comparison with the work in [14,15], it exhibits pure capacitive filtering at the output, which enables a classical PI voltage control method to be used. Additionally, with three voltage level characteristics reached at the output in conjunction with the asymmetrical/independent operation of the integrated boost structures, the proposed converter topology could also be used in voltage balancing or interconnection with bipolar DC grids/applications [32]. Another major contribution of the present work is the unique integration of a multitude of useful features, such as interleaved, high-gain, good efficiency, low voltage stress, three-level output with symmetric/asymmetric operation, and stable control method, into a single electronic conversion solution.

Apart from the introduction, the paper is organized in the following structure. In Section 2, a detailed representation of the proposed topology and switching states is presented. Furthermore, an analysis of the converter in both the continuous conduction mode and discontinuous conduction mode is discussed in detail. Section 3 presents a study of the converter's behavior, performed by software simulation and practical measurements on a laboratory model. In addition, the main features of the proposed topology in comparison with similar approaches are emphasized in this section. The conclusions and further work are presented in Section 4.

## 2. Converter Topology Analysis

### 2.1. Converter Topology, Switching States, and Presumptive Waveforms

The proposed converter topology, named an independent double-boost interleaved converter and henceforth referred to as IDBIC, is based on a patent application [33] developed especially for PV/wind energy harvesting and battery energy conversion systems. The basic converter topology is represented in Figure 1, while the main operating stages of the converter are presented in Figure 2, in which nine independent switching states (S1–S9) are underlined.

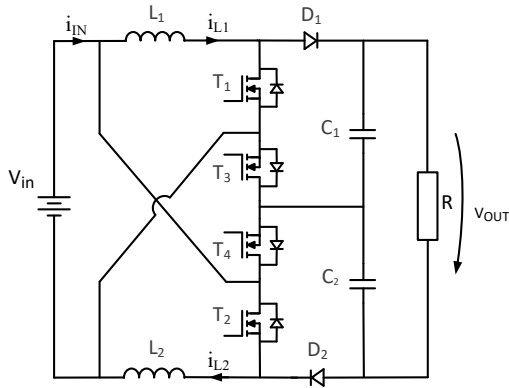

**Figure 1.** The electronic schematic of the proposed converter—IDBIC.

S1—Charging L₁; Discharging L₂.

S2—Discharging L₁; Discharging L₂.

S3—Discharging L₁; Discharging L₂.

S4—Charging L₁; Charging L₂.

S5—Discharging L₁; Charging L₂.

S6—Discharging L₂.

S7—Discharging L₁.

S8—Charging L₁.

S9—Charging L₂.

**Figure 2.** Switching stages for the proposed IDBIC converter.

The characteristic waveforms during operation are introduced in Figure 3, in which the continuous conduction mode (CCM) and discontinuous conduction mode (DCM) operations are exemplified in conjunction with the S1–S9 switching states. These four presumptive switching patterns are mainly triggered by the duty cycle values of the T1 and T2 transistor command signals. In view of this, for a duty cycle smaller than 0.5, the inductor currents have steeper falling slopes during the S2 and S3 switching stages. For a duty cycle larger than 0.5, inductor current waveforms and behavior similar to the regular boost interleaved converter can be observed.

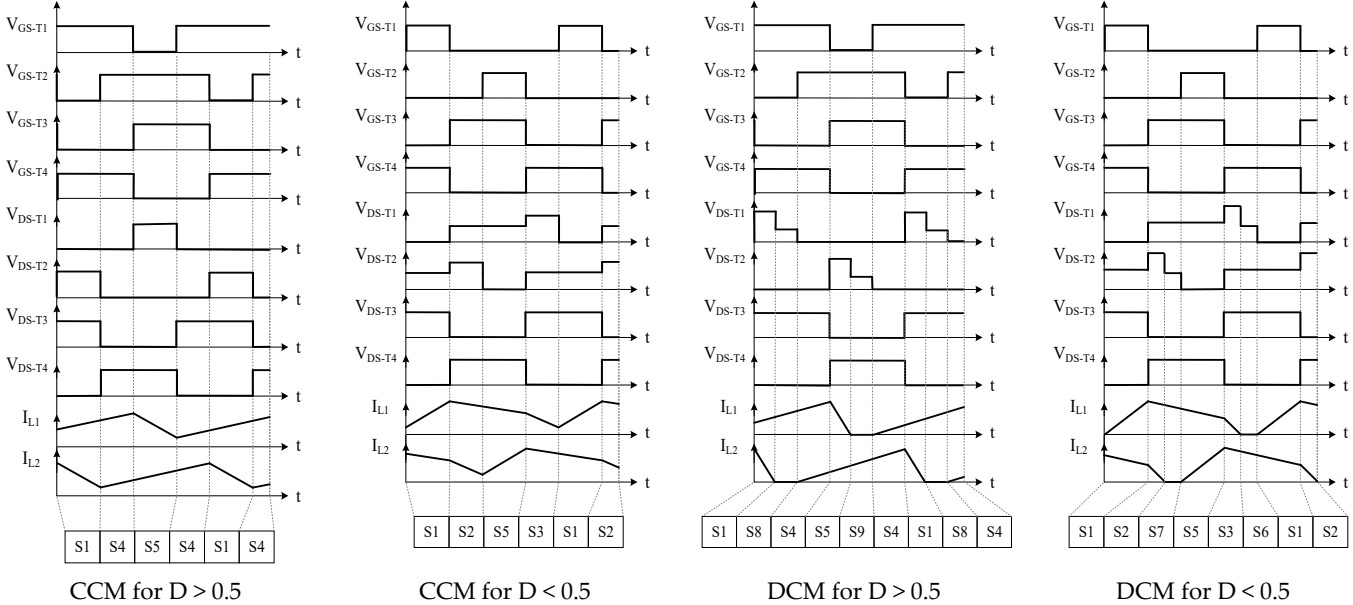

| CCM for D > 0.5 | CCM for D < 0.5 | DCM for D > 0.5 | DCM for D < 0.5 |

$V_{GS-T1}$, $V_{GS-T2}$, $V_{GS-T3}$, $V_{GS-T4}$—gate-source voltage for $T_1$–$T_4$ transistors; $V_{DS-T1}$, $V_{DS-T2}$, $V_{DS-T3}$, $V_{DS-T4}$—drain-source voltage for $T_1$–$T_4$ transistors; $I_{L1}$, $I_{L2}$—$L_1$ and $L_2$ inductor currents; D—duty cycle; CCM—continuous conduction mode; DCM—discontinuous conduction mode; S1–S8—switching stages from Figure 2.

**Figure 3.** Presumptive functioning waveforms and switching stage correlations.

As documented in these switching stages and presumptive functioning patterns, the two integrated boost converters work independently from each other, with the inputs connected in parallel and the outputs in series. From this observation, one of the key characteristics of the proposed double-boost converter is defined.

### 2.2. CCM Operation of the Proposed Converter

Considering the CCM operation, Figure 4 shows the steady-state waveforms for the inductor voltage and current at a duty cycle D larger and smaller than 0.5. For a duty cycle D larger than 0.5, the voltage ratio of the proposed converter can easily be deduced by the classic boost converter analysis approach [34], where the average inductor voltage equation is:

$$V_{in}D + (V_{in} - V_{C1})(1 - D) = 0 \tag{1}$$

where the capacitors $C_1$ and $C_2$ voltages are:

$$V_{C1} = V_{C2} = \frac{V_{out}}{2} \tag{2}$$

The deduced voltage gain is given as:

$$\frac{V_{out}}{V_{in}} = \frac{2}{(1 - D)} \tag{3}$$

Working at a duty cycle D smaller than 0.5, the average inductor voltage equation is:

$$V_{in}D + (V_{in} - V_{C1})0.5 - V_{C1}(0.5 - D) = 0 \tag{4}$$

Thus, for this condition, the voltage gain can be expressed as:

$$\frac{V_{out}}{V_{in}} = \frac{2D + 1}{(1 - D)} \tag{5}$$

Considering (3) and (5), Figure 5 shows the DC conversion ratio M(D), of the proposed IDBIC converter for CCM operation in conjunction with the regular boost converter gain.

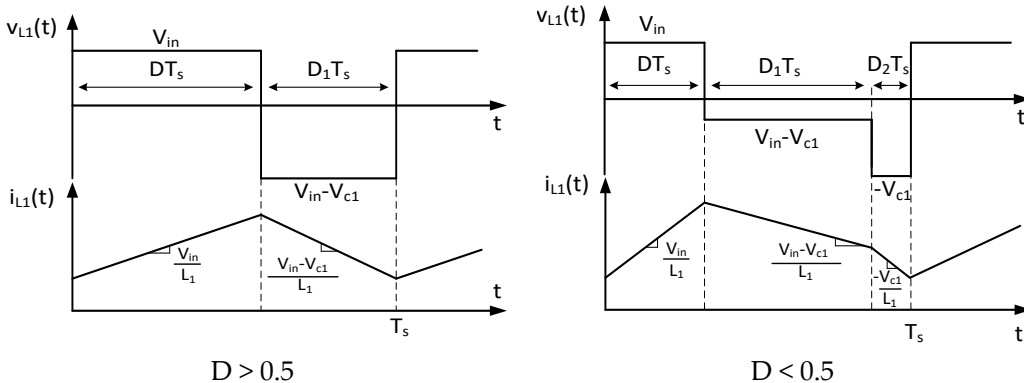

**Figure 4.** CCM inductor voltage and current waveforms.

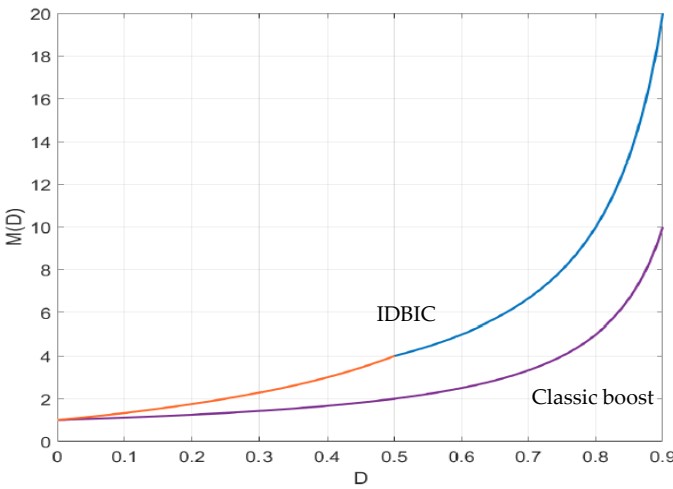

**Figure 5.** Classic boost converter (purple line) and IDBIC (for D < 0.5 the orange line and for D > 0.5 the blue line) DC conversion ratio M(D) in CCM.

### 2.3. DCM Operation of the Proposed Converter

2.3.1. Case 1

Figure 6 shows the steady-state waveforms of inductor voltage and current for DCM operation in Case 1, where three different time intervals (D, $D_1$, and $D_2$) can be observed in a switching period. These waveforms are specific to the regular DCM boost converter [34].

From the average inductor voltage equation, the time interval defined by the time $D_1$ in Figure 6 can be expressed as:

$$D_1 = \frac{2V_{in}D}{V_{out} - 2V_{in}} \tag{6}$$

Considering the output C1 capacitor charge balance, the converter load resistance R, the current area A from Figure 6, and the peak current $i_{pk}$, the diode D1 average current equation can be described as:

$$\langle i_{D1} \rangle = \frac{1}{T_s} A = \frac{1}{T_s} \left( \frac{1}{2} i_{pk} D_1 T_s \right) = \frac{V_{in} D D_1 T_s}{2L_1} = \frac{2V_{C1}}{R} \tag{7}$$

By substituting the time interval $D_1$ into (7), the voltage gain in DCM Case 1 becomes:

$$\frac{V_{out}}{V_{in}} = \frac{\left( 2 + \sqrt{4 + \frac{8D^2}{K}} \right)}{2} \tag{8}$$

where:

$$K = \frac{2L_1}{RT_s} \tag{9}$$

Using (8) and different values of the parameter K, Figure 7 shows the DC conversion ratio, M(D,K), of the IDBIC boost converter for DCM Case 1.

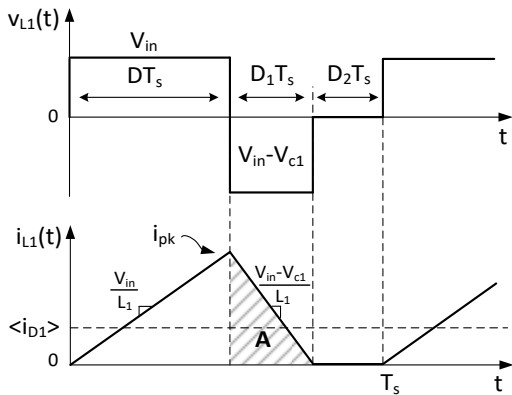

**Figure 6.** Inductor voltage and current steady-state waveforms—DCM Case 1.

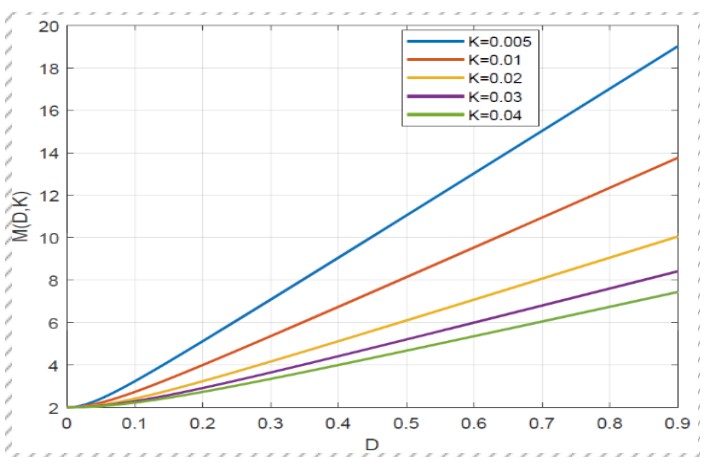

**Figure 7.** DC conversion ratio M (D, K) of the boost converter—DCM Case 1.

### 2.3.2. Case 2

This DCM working case is illustrated in Figure 8, in which the converter is working at a duty cycle smaller than 0.5, where four different time intervals (D, $D_1$, $D_2$, and $D_3$) can be observed, and the inductor current has two falling slopes. For this operation mode, the average current of the diode D1 can be expressed using the current represented by the area A and the output capacitor $C_1$ charge balance.

Using the average inductor voltage equation, the expression of the time interval $D_2$ defined in Figure 8 is obtained:

$$D_2 = \frac{V_{in}}{V_{C1}} (D + D_1) - D_1 \tag{10}$$

Using the output $C_1$ capacitor charge balance, the converter load resistance R, and the current area A from Figure 8, the diode $D_1$ average current equation can be expressed as:

$$\langle i_{D1} \rangle = \frac{2V_{C1}}{R} = \frac{1}{T_S} A \tag{11}$$

By substituting the time interval defined by $D_2$ into the capacitor $C_1$ charge balance (11), the result is:

$$2KV_{C1}^2 + V_{C1}V_{in}D_1^2 - V_{in}(D + D_1) = 0 \tag{12}$$

Knowing that the time interval $D_1$ is 0.5, the voltage ratio of the converter becomes:

$$\frac{V_{out}}{V_{in}} = \frac{-0.5^2 + \sqrt{0.5^4 + 8K(D + 0.5)^2}}{K} \tag{13}$$

Using (13) and different values of the parameter K, Figure 9 shows the DC conversion ratio, M(D, K), of the IDBIC boost converter for DCM Case 2.

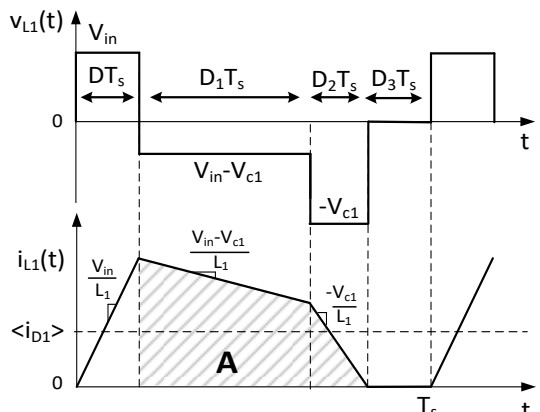

**Figure 8.** Inductor voltage and current steady-state waveforms—DCM Case 2.

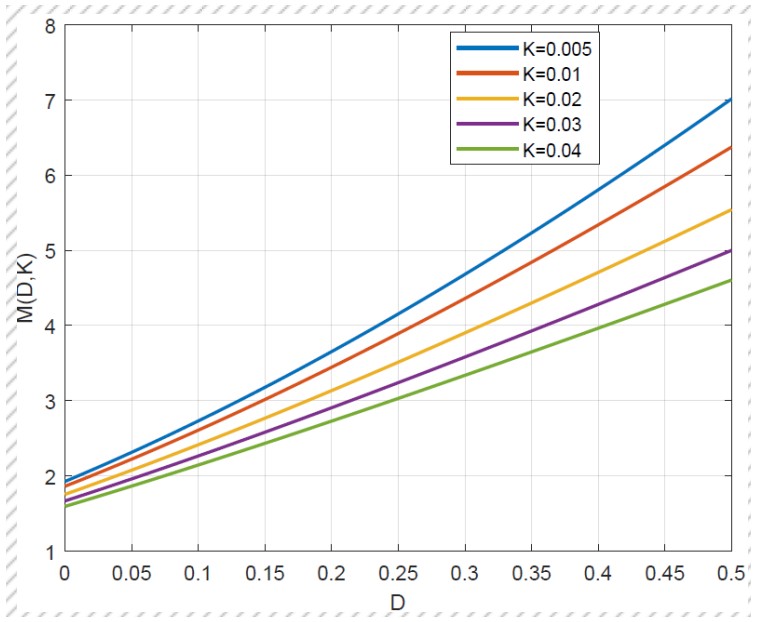

**Figure 9.** DC conversion ratio M (D, K) of the boost converter—DCM Case 2.

*2.4. DCM–CCM Boudary Limit*

In Figure 10, the DCM–CCM boundary limit is presented. Using the voltage gain Equations (3) and (8), the plot was obtained for D larger than 0.5, which complies with (14). For D smaller than 0.5, Equations (5) and (13) were used, and the evolution of the parameter $K_{crt}$ is defined by (15).

$$K_{crt}(D) = \frac{D(1-D)^2}{2} \tag{14}$$

$$K_{crt}(D) = \frac{(1-D)\left(1 + 2D - 4D^2\right)}{4(2D+1)} \tag{15}$$

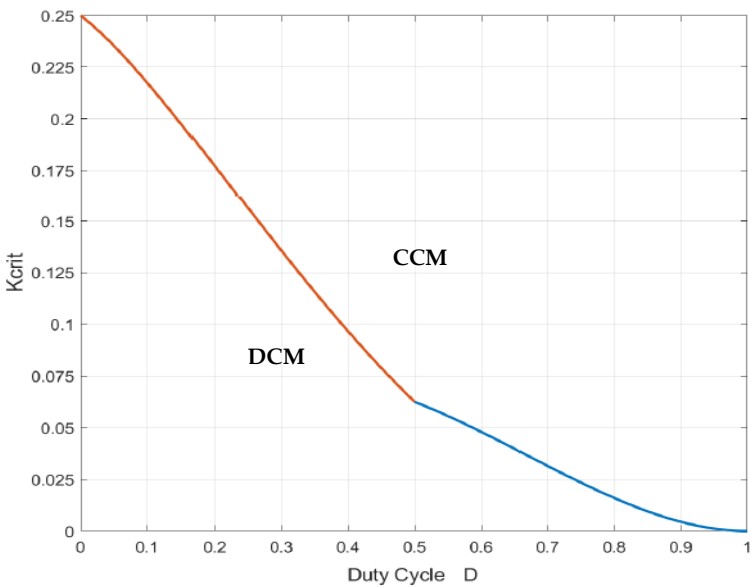

**Figure 10.** CCM–DCM boundary limit plot for D < 0.5 (orange line) and for D > 0.5 (blue line).

## 3. Simulation and Practical Implementation

Figures 11 and 12 introduce the PLECS simulation and practical measurements for the discontinuous conduction mode (DCM) and continuous conduction mode (CCM) of the proposed IDBIC converter in the symmetrical operation of the two integrated boost converters. The gate-source command voltages of the transistors ($T_1$, $T_2$, $T_3$, $T_4$) are represented by $V_{GS\text{-}T1}$, $V_{GS\text{-}T2}$, $V_{GS\text{-}T3}$, and $V_{GS\text{-}T4}$. The drain-source voltages on all transistors are shown as $V_{DS\text{-}T1}$, $V_{DS\text{-}T2}$, $V_{DS\text{-}T3}$, and $V_{DS\text{-}T4}$. The inductors' currents, $I_{L1}$ and $I_{L2}$, are illustrated as well. The command signals $V_{GS\text{-}T1}$ and $V_{GS\text{-}T2}$ are pulse width modulation signals (PWM), normally used in the regular interleaved boost converter with a 180° phase shift between the two signals. The logic 1 (high value) of the gate signal $V_{GS\text{-}T3}$ was obtained between the falling edges of the $V_{GS\text{-}T1}$ and $V_{GS\text{-}T2}$ signals. The gate signal $V_{GS\text{-}T4}$ was obtained by inverting the signal $V_{GS\text{-}T3}$. A compulsory deadtime is introduced for the $T_3$ and $T_4$ transistor gate signals.

As one can notice, no significant difference can be observed between the simulated and practical measurements. The main differences are obtained during the DCM modes when the inductors' currents are zero. Because of the parasitic components associated with the switching devices and PCB layout, resonant and transient behavior can be observed during the practical measurements. Despite this phenomenon, it is important to note that no relevant negative effect was present in the working behavior of the converter.

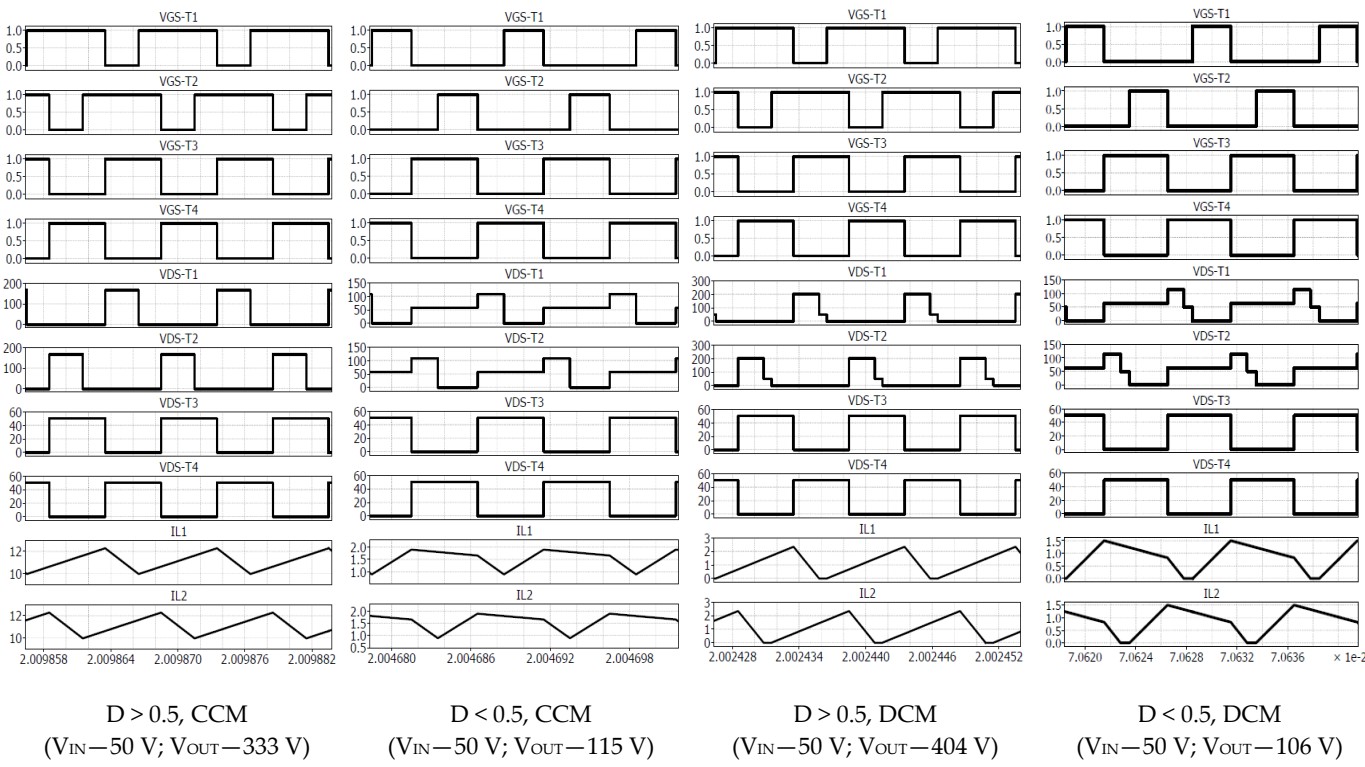

**Figure 11.** PLECS Simulation results in DCM and CCM operation modes.

D > 0.5, CCM
(V$_{IN}$—50 V; V$_{OUT}$—333 V)

D < 0.5, CCM
(V$_{IN}$—50 V; V$_{OUT}$—115 V)

D > 0.5, DCM
(V$_{IN}$—50 V; V$_{OUT}$—404 V)

D < 0.5, DCM
(V$_{IN}$—50 V; V$_{OUT}$—106 V)

D > 0.5, CCM
(V$_{IN}$—50 V; V$_{OUT}$—322 V)

D < 0.5, CCM
(V$_{IN}$—50 V; V$_{OUT}$—142 V)

D > 0.5, DCM
(V$_{IN}$—50 V; V$_{OUT}$—417 V)

D < 0.5, DCM
(V$_{IN}$—50 V; V$_{OUT}$—131 V)

**Figure 12.** Practical measurements of the proposed converter in DCM and CCM.

To operate the integrated boost converters in an independent manner, a three-voltage level system with two loads is needed at the output, as one can notice in Figure 13, where the generic voltage control loops and the PWM signal generator are highlighted. This schematic is suitable for an asymmetric control of the converter in which the reference voltages $V_{ref\,1}$ and $V_{ref\,2}$ can have different values; thus, each integrated boost converter needs to operate independently.

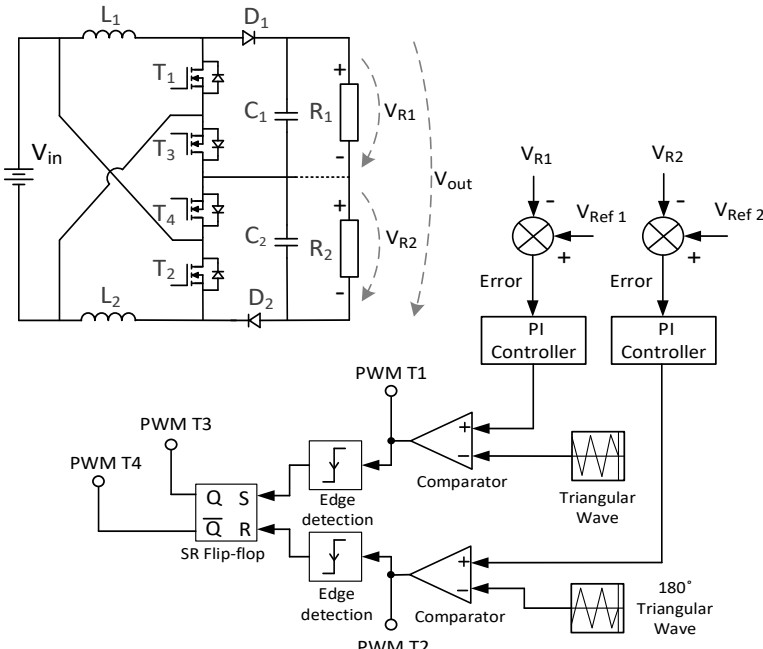

**Figure 13.** The generic output voltage control loop and PWM generator.

The most common usage of the asymmetrical control is related to the energy balancing/interconnections in bipolar DC grids/applications. Thus, in this case, the voltage references $V_{ref\,1}$ and $V_{ref\,2}$ are equal. Considering different values for the $R_1$ and $R_2$ loads in the asymmetric mode, indicating different inductor currents, the independent operation of the two integrated boost converters is demonstrated. For an application in which the overall output voltage $V_{out}$ must be regulated, a symmetrical control loop approach is sufficient, meaning that only one PI controller and one voltage reference $V_{ref}$ are needed.

Figure 14 presents the practical measurements for the symmetric and asymmetric control of the converter considering the control schematics presented in Figure 13. For this situation, the output voltage $V_{out}$ is set at 400 V, and the voltages $V_{R1}$ and $V_{R2}$ are equal to 200 V. During CCM operation, the command signals are almost identical, while the inductor currents are equal for the symmetric control and have different values for the asymmetrical control.

Moreover, based on an application with 50 V and 100 V input voltages and a 400 V output DC voltage in symmetric operation mode, the efficiency measurements carried out with the Tektronix PA3000 power analyzer are illustrated in Figure 15. The laboratory test setup is presented in Figure 16. For the prototype, the components and some general test specifications are presented in Table 1. In addition, Table 2 comprises the generalized information regarding the main characteristics of the proposed converter, directly compared with actual non-isolated topologies usually found targeting the same applications.

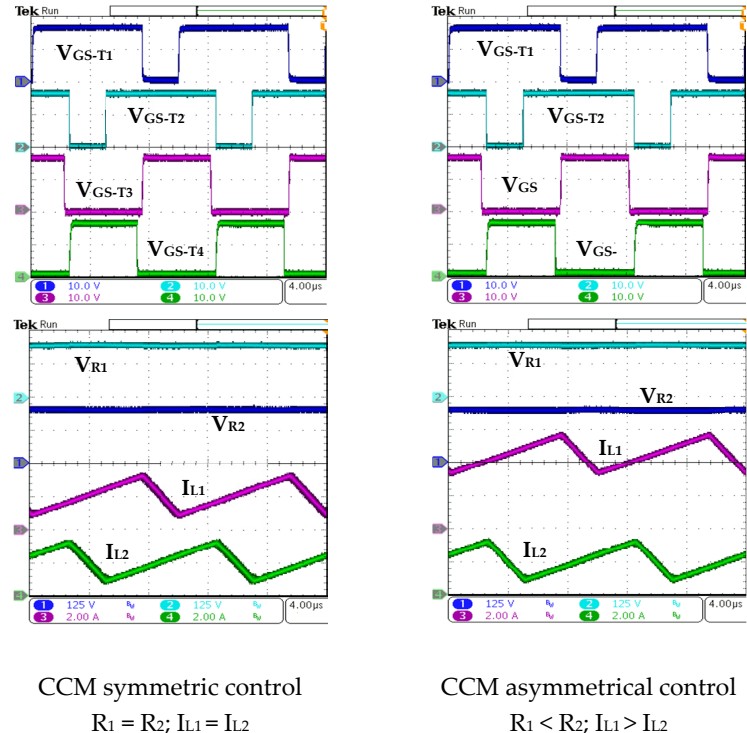

CCM symmetric control
R₁ = R₂; I_{L1} = I_{L2}

CCM asymmetrical control
R₁ < R₂; I_{L1} > I_{L2}

**Figure 14.** Practical measurements of the proposed converter in symmetrical and asymmetrical operation modes.

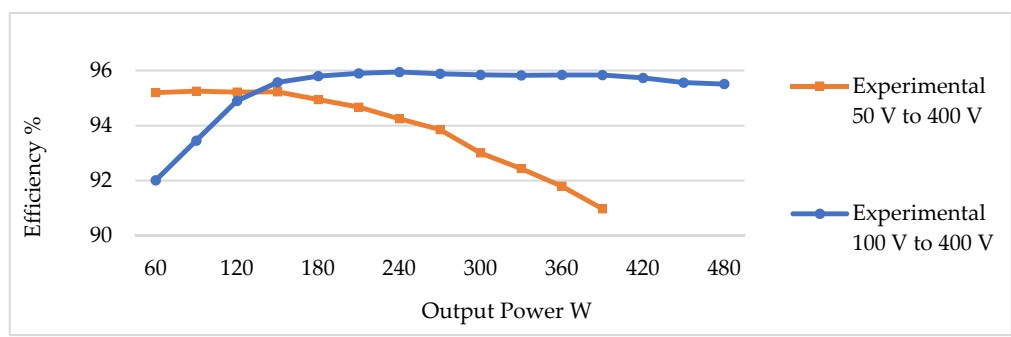

**Figure 15.** Laboratory practical efficiency measurements.

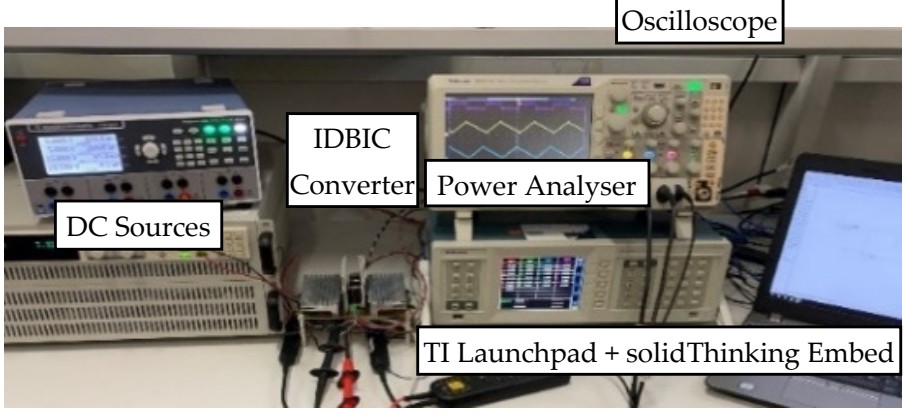

**Figure 16.** Laboratory test setup.

**Table 1.** Prototype components and specifications.

| Parameters | Values |
|---|---|
| Input Voltage—$V_{IN}$ | 50 V–100 V |
| Output Voltage—$V_{OUT}$ | 400 V |
| Switching frequency—$f_s$ | 100 kHz |
| Max. Output Power—$P_{OUT}$ | 480 W |
| Switches ($T_1$–$T_4$) | C2M0280120D |
| Diodes (D1, D2) | C3D02060A |
| Inductors ($L_1$, $L_2$) | 50 µH, RL = 21 mΩ |
| Capacitors ($C_1$, $C_2$) | 100 µF |

**Table 2.** Converter characteristics comparison.

| Ref. | Voltage Gain (M) | Normalized Voltage Stress across the Power | | $P_{out}$ [W] | Efficiency [%] at $V_{in}$ [V] | Components S*/D*/L*/ C*/C.I*/T* |
|---|---|---|---|---|---|---|
| | | Switches $V_S/V_O$ | Diodes $V_D/V_O$ | | | |
| IDBIC | $2/(1-d)$ | $1/M+0.5$ $1/M$ | 0.5 | 240 | 94.25 @ 50 V 95.95 @100 V | 4/2/2/ 2/-/10 |
| [15] | $(1+d)/(1-d)$ | $(M+1)/2M$ | $(M+1)/2M$ | 240 | 91.7 @ 24 V | 2/2/2/ 4/2/10 |
| [17] | $1/(1-d)$ | 0.5 | 0.5 | 300 | 95.9 @ 60 V | 4/2/2/ 3/-/11 |
| [18] | $2/(1-d)$ | 0.5 | 0.5 | 320 | 90.2 @ 48 V 95 @ 80 V | 2/3/2/ 3/-/10 |
| [20] | $(3+d1-d2)/(1-d1-d2)$ | $(M+1)/4M$ $(M-1)/2M$ | $(M+1)2M$ | 500 | 93.4 @ 20 V 95.85 @ 30 V | 3/4/2/ 3/-/12 |
| [21] | $(1+5d)/(1-d)$ | $(1+5M)/6M$ | $(M+1)/M$ | 200 | 95.9 @ 30 V | 6/9/6/ 1/-/22 |
| [22] | $2/(1-d)$ | 0.5 | - | 300 | 91.3 @ 50 V 94.3 @ 100 V | 4/0/1/ 4/0/9 |
| [23] | $3/(1-d)$ | 0.33 | 0.33 | 150 | 93.9 @30 V | 1/5/1/ 5/0/12 |
| [26] | $(3+d)/(1-d)$ | $(M+1)/4M$ | $(M+1)/2M$ | 200 | 96 @ 30 V | 2/3/-/ 3/1/9 |

S*: switch, D*: diode, L*: inductor, C*: capacitor, C.I*: coupled inductor, T*: total of components, $V_O$: output voltage, $V_S$: switch voltage, $V_D$: diode voltage, $V_{in}$: input voltage, $P_{out}$: output power, d: duty cycle.

For analyses in wide voltage range applications, the laboratory prototype testing setup from Figure 16 was developed using high-voltage transistors, which exhibit quite high internal resistances (0.23 Ω). As one can see in Figure 15, at a 50 V input voltage, the efficiency decreases at high powers because of the transistors' conduction losses that become predominant.

## 4. Conclusions

This work emphasizes the development and analysis of an interleaved converter with independent operation of the two integrated boost converter stages, attaining at the output a three-level voltage structure. The symmetric and asymmetric operations were demonstrated, together with all possible CCM and DCM operations.

Although the prototype design is not fully optimized, the results are encouraging, and future improvements can be aimed at the selection of the electronic switching devices, analog components, and PCB design. The converter features good energy efficiency with twice the voltage gain of regular boost and interleaved topologies.

The proposed converter has no specific feature with outstanding performance, and in certain applications, a shortcoming of the converter could be that the output and input do not share a common ground. Moreover, the complexity of the proposed structure can be considered a drawback, but by increasing the number of power electronics applications and particularities, alternative approaches will increase as well (Table 2). Compared with these solutions, the general number of components in the proposed converter is not very high. Nevertheless, considering all the combined characteristics, namely the interleaved property, high gain, good efficiency, low voltage stress, output with three-voltage levels, and especially the independent control of the integrated electronic structures, this converter is an engaging solution for PV optimizers/microinverters and battery energy management

systems. Hence, thorough analytical and experimental studies are foreseen in future work to validate the proposed structure in DC grid interconnections and other applications by finding the best performance and optimal command/control strategies.

**Author Contributions:** Conceptualization, V.M.S. and P.D.T.; formal analysis, S.I.S.; investigation, V.M.S., S.I.S. and A.M.P.; methodology, V.M.S. and P.D.T.; software, V.M.S. and L.N.P.; supervision, P.D.T.; validation, V.M.S., A.M.P. and N.C.S.; writing—original draft, V.M.S., S.I.S. and P.D.T.; writing—review and editing, V.M.S., S.I.S. and P.D.T. All authors have read and agreed to the published version of the manuscript.

**Funding:** This research was funded by the European Regional Development Fund through the Competitiveness Operational Program 2014–2020 Romania, grant number 16/01.09.2016, project title "High power density and high-efficiency micro-inverters for renewable energy sources—MICROINV.

**Informed Consent Statement:** Not applicable.

**Data Availability Statement:** Not applicable.

**Conflicts of Interest:** The authors declare no conflict of interest.

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
