# Peer review of "Independent Double-Boost Interleaved Converter with Three-Level Output"

_applsci, doi:10.3390/app11135993_

Round 1

Reviewer 1 Report

This paper describes a novel converter architecture based on an independent controlled double boost topology. The novel topology is firstly illustrated, to later develop the mathematical model. Finally, some experimental results are obtained on a benchmark testbed to prove the effectiveness of the developed device. In general, the paper is not well-written. An extensive revision of the English is needed and more references should be added. Contributions are neither described nor justified. Despite these weaknesses, the topic covered in this work is on interest and perfectly match with the scope of this special issue. This way, it deserves a reconsideration after major modifications. More specifically, my comments are enumerated below:

  • The whole text should be revisited to improve the quality of English. For instance, the authors repeatedly use the word ‘obtained’ in the Abstract. I suggest using synonyms to improve the readability of the manuscript. Other example is the word ‘acknowledged’ in the second paragraph of the Introduction, maybe the phrase could be reconstructed. Similar examples can be encountered through the text.
  • In the very first phrase of the article, the authors stated that ‘The need for high gain boost topology converters with low ripple of the input current could be justified in various applications’. Such as it is stated in the text, this phrase is very few justified. The authors should include either references or examples to support this statement. One example could be the deployment of bidirectional chargers for electric vehicles in smart homes, which enable vehicle-to-home capability of the on-board system allowing energy flows from and to the home. In this sense, I suggest including the reference [A].

[A]          M. Tostado-Véliz, R. S. León-Japa, F. Jurado. Optimal electrification of off-grid smart homes considering flexible demand and vehicle-to-home capabilities. Applied Energy 2021; 298: 117184. https://doi.org/10.1016/j.apenergy.2021.117184

  • In general, the Introduction is poor in the sense that this Section does not well explain the motivation and contributions of the paper. I suggest revisiting this section, emphasizing the major contributions of the article that justify its publication.
  • The Literature Review should be completed with a summary table of similar architectures existing in the literature, highlighting the main differences of the novel converter w.r.t. to other devices.
  • The article layout should be described at the end of the Introduction, as customary.
  • Acronyms should be described the first time that they are used in the text. For example, in page 2 lines 79-80, the acronym IDBIC is not well introduced.
  • The authors point out that one of the main potential applications of the introduced device is the interconnection with DC grids. In this sense, I suggest carrying out more experiments in this direction. I know that maybe this task could be complex for a first stage on this research line, but at least the authors should comment this as a main future research line.

Many thanks to the authors for your effort and time.

Author Response

Dear reviewer,

We would like to thank you for the useful comments and discussions given to improve the quality of the paper and hopefully, our answers are accordantly to your expectations.

Our responses are:

  1. We have updated the whole text to fulfill the useful reviewer recommendations (all the changes are highlighted using Track Changes in Word);
  2. We have updated the introduction with 8 new references to highlight different target applications (page 1, rows 27-29);
  3. In the introduction section we have inserted several new statements regarding the motivation, novelty, and the main contributions of the paper (ex. page 2, rows 70-81);
  4. Regarding the comparison with similar converter topologies, this has been done in Table 2, page 12;
  5. A summary of the paper has been inserted at the end of the Introduction chapter (page 2, rows 82-89);
  6. We have updated the acronym IDBIC introduction (page 2, rows 92-95). Also, other acronyms used in the paper were properly defined/redefined;
  7. As anticipated by the reviewer, it is quite difficult to perform DC microgrids testing in a short time using the proposed converter, although we are working intensively in this research field. In view of this, we have inserted this latter information in the conclusions section (page 13, rows 265-268).

Thank you again for the helpful inputs!

Kind regards,

The authors

Reviewer 2 Report

This paper develops a new DC/DC converter topology based on an independent controlled double boost topology. It combines two classic boost converters connected in parallel at the input and in series at the output. An interleaved topology with a low input current ripple is obtained by controlling the two boost converters with different time phases. It has appealing features, such as the possibility of a three-voltage level structure with a high gain compared to classic boost and interleaved topologies. The paper details the operation principle, simulation results, a mathematical analysis as well as  experimental results.

The work is valuable since it falls in a very interesting research area, although some aspects can be improved as detailed below.

1.- Introduction section. The contributions and novelties of the paper with respect to the state-of-the-art require more details.

2.- Figure 15. The 50V-400 V and 100V-400 V efficiency curves show opposite trends. While the efficiency of the first curve decreases when increasing the output power, the second one increases and stabilizes with the output power. The authors must explain the origin of this divergence.

3.- The authors should compare the complexity of this solution in terms of components required, switches and controllability with respect to similar solutions.

4.- It would be nice to include the critical points of this converter, if any, compared with those of similar solutions.

The Reviewer aims the authors to revise the work based on the suggestions above in order to improve its quality

Author Response

Dear reviewer,

We would like to thank you for the useful comments and discussions given to improve the quality of the paper and hopefully, our answers are accordantly to your expectations.

Our responses are:

  1. We have updated the introduction with 8 new references to highlight different target applications (page 1, rows 27-29); More, we have inserted several new statements regarding the motivation, novelty, and the main contributions of the paper (ex. page 2, rows 70-81); A direct comparison of the main features of the converter are presented in Table 2, page 12;
  2. Thank you for your useful comment! We have updated the paper explaining the slope of the efficiency at low input currents (page 11, rows 238-243);
  3. Regarding the comparison of the complexity of the solution with similar converter topologies, this has been done in Table 2, page 12;
  4. Two important drawbacks can arise from using the proposed topology. One is the complexity of the structure, namely the number of components, and the second the fact that the load and source do not share a common ground. The latter drawback is addressed in the paper on page 12, rows 259-261. The issue regarding the complexity of the structure in an ideal approach, fewer components are always better. In real applications one can face certain limitations and, in some cases, the right performances cannot be achieved with the simple, classic, and well-established topologies. More, the growing number of power electronics applications and particularities, alternative approaches are increasing as well (table 2). Comparing with these solutions, the general number of components in the present work is not that high, also considering that some key advantages are united in one converter topology. We have inserted this information in the conclusion chapter, page 12-13, rows 261-264)

Thank you again for the helpful inputs!

Kind regards,

The authors

Round 2

Reviewer 1 Report

I have not further comments.

Congratulations to the authors.